# ReSTiNet: On Improving the Performance of Tiny-YOLO-Based CNN Architecture for Applications in Human Detection

**Shahriar Shakir Sumit** [1,*]**, Dayang Rohaya Awang Rambli** [1]**, Seyedali Mirjalili** [2,3]**, Muhammad Mudassir Ejaz** [4] **and M. Saef Ullah Miah** [5]

1. Department of Computer & Information Sciences, Universiti Teknologi PETRONAS (UTP), Seri Iskandar 32610, Perak, Malaysia
2. Centre for Artificial Intelligence Research and Optimization, Torrens University Australia, Brisbane, QLD 4006, Australia
3. Yonsei Frontier Lab, Yonsei University, 50 Yonsei-ro Seodaemun-gu, Seoul 03722, Korea
4. Electrical & Electronics Engineering, Universiti Teknologi PETRONAS (UTP), Seri Iskandar 32610, Perak, Malaysia
5. Faculty of Computing, College of Computing and Applied Sciences, Universiti Malaysia Pahang, Pekan 26600, Pahang, Malaysia
* Correspondence: shahriar9121@gmail.com

**Abstract:** Human detection is a special application of object recognition and is considered one of the greatest challenges in computer vision. It is the starting point of a number of applications, including public safety and security surveillance around the world. Human detection technologies have advanced significantly in recent years due to the rapid development of deep learning techniques. Despite recent advances, we still need to adopt the best network-design practices that enable compact sizes, deep designs, and fast training times while maintaining high accuracies. In this article, we propose ReSTiNet, a novel compressed convolutional neural network that addresses the issues of size, detection speed, and accuracy. Following SqueezeNet, ReSTiNet adopts the fire modules by examining the number of fire modules and their placement within the model to reduce the number of parameters and thus the model size. The residual connections within the fire modules in ReSTiNet are interpolated and finely constructed to improve feature propagation and ensure the largest possible information flow in the model, with the goal of further improving the proposed ReSTiNet in terms of detection speed and accuracy. The proposed algorithm downsizes the previously popular Tiny-YOLO model and improves the following features: (1) faster detection speed; (2) compact model size; (3) solving the overfitting problems; and (4) superior performance than other lightweight models such as MobileNet and SqueezeNet in terms of mAP. The proposed model was trained and tested using MS COCO and Pascal VOC datasets. The resulting ReSTiNet model is 10.7 MB in size (almost five times smaller than Tiny-YOLO), but it achieves an mAP of 63.74% on PASCAL VOC and 27.3% on MS COCO datasets using Tesla k80 GPU.

**Keywords:** computer vision; object detection; human detection; convolutional neural networks

## 1. Introduction

Human beings possess an inherent ability to perceive surrounding objects in static images or image sequences almost flawlessly. They can also sense emotions and interactions among persons and notice the total persons present in images by making mere observations. The computer vision field is expected to provide the required technological assistance for this human aptitude in order to improve the quality of life of humans. Hence, the aim of this field is to explore methods for effectively teaching machines or computers to observe and understand characteristics in images or videos using digital cameras [1].

A precise detection of objects in an image is essential in computer vision in order to suit the demands of various applications involving vision-based approaches. For instance,

object detection includes the identification of specific details in an image, and localizing its coordinates is considered to be a problem in vision technology. Identifying objects is not the only task that requires performance but categorizing them accordingly across various classes in an appropriate manner is also required [2]. A classic example of this includes visual object detection [2]. Figure 1 illustrates the basic operation of a machine learning (ML) model for detecting objects. For example, consider the goal of classifying three dissimilar objects: a bird, a human being, and a lion. Initially, training images are collected with labeled data in preparation for training an ML framework. Secondly, the desired features are extracted and then added to the classifier's architecture.

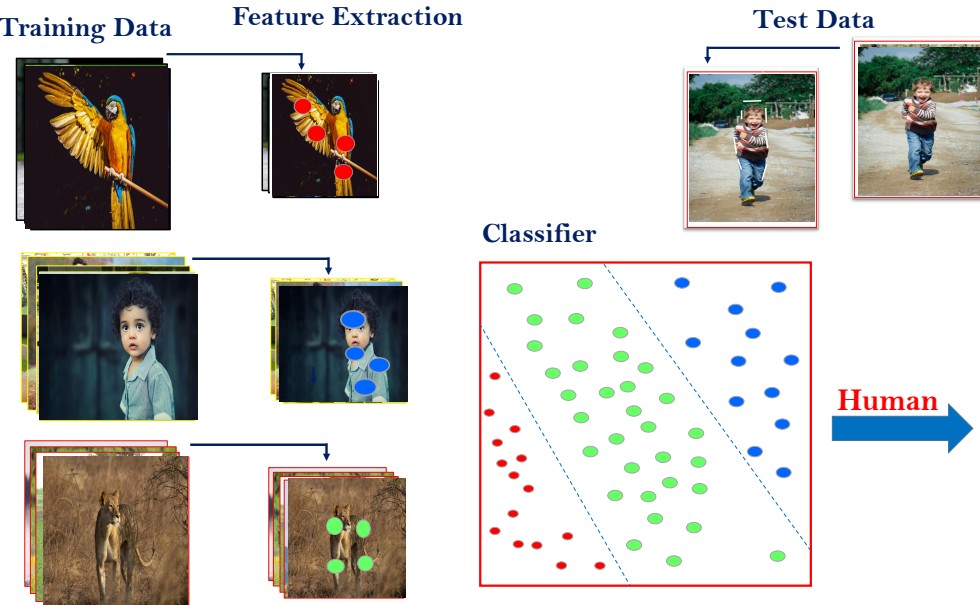

**Figure 1.** Example of machine learning work flow for object classification across three different classes (bird, human, and lion).

Certain features can be best expressed by utilizing various object characteristics that include colors, corners, edges, ridges, and regions or blobs [3]. The success achieved from training is directly proportional to several factors that include feature extraction, classifier selection, and the training procedure. The first task is important as it not only enhances the accuracy of trained networks but also eliminates redundant features in the image. It involves reducing the dimensionality of data by extracting redundant information, which in turn improves the quality of inference while simultaneously improving the training rate. An ideological view is to expect features to be invariable in the control of dynamic and illuminated conditions while possessing the capability to cope with any randomized variations during either scaling or rotational motions. Features are appended to the training framework after all feasible features from the image samples have been extracted. They are then supplied to an appropriate sort of classifier based on accuracy and speed. Some normally used classifiers exist, which include the Support Vector Machines (SVM), Nearest Neighbor (NN), Random Forest (RF), and Decision Tree (DT). Once the training framework is ready, removing alike features from the test image samples and, as a consequence, predicting the proper class from features using the trained framework for each provided test image are feasible.

Several techniques were proposed in light of the efficient extraction of features as well as classification to detect arbitrary objects in images [4]. Over the past two decades, the focus had been on the design of efficient hand-crafted features to improve detection robustness and accuracy. A diverse set of extraction techniques was provided by the vision research community such as Scale Invariant Feature Transform (SIFT), Viola Jones

(VJ), Histogram of Oriented Gradients (HoG), Speeded Up Robust Features (SURF), and Deformable Part-Based Models (DPM) [5,6].

Deep learning techniques have effectively combined the task of extracting the features and classification in an end-to-end way [4]. Convolutional neural networks (CNNs) have become quite popular for tackling various problems, among which includes object detection. Subsequently, the performance of such architectures has led to a proliferation in both achievable speed and accuracy. The object detection methods using deep CNN such as Spatial Pyramid Pooling Networks (SPPNets), Region-based CNN (R-CNN), Feature Pyramid Networks (FPNs), fast RCNN, You Only Look Once (YOLO), faster R-CNN, Single-Shot Multibox Detector (SSD), and Region-based Fully Convolutional Networks (R-FCNs) have shown excellent benefits relative to state-of-the-art ML methods [7,8]. This article focuses on a specific sub-domain of detection, which is the human detection.

In a year, over a billion people lost their lives and around 20–50 million people experienced fatal complications as a result of traffic accidents [9]. In 2015, more than 5000 pedestrians died in traffic accidents, while about 130,000 pedestrians required medical care for non-fatal problems in the United States. However, the ratio of traffic fatality can be reduced or even eliminated by utilizing various detection techniques in autonomous vehicles that use sensors to interact with other neighboring vehicles in the vicinity [10].

With increases in crime and public fear of terrorism, public security has become an unavoidable concern, and human detection techniques can be employed to monitor and control public spaces remotely. Approximately 21,000 people lose their lives because of terrorist activities every year and 0.05% of the total deaths in 2017 occurred due to terrorism [11]. The necessity to install a sufficient number of human-detecting devices has spiked in public locations following tragedies in London, New York, and other cities across the globe. Such incidents are critical enough and demand a robust design and global deployment of such systems. Hence, human-detection systems are observed as a viable answer for ensuring public safety and have become one of the most significant study fields today.

The detection of human beings is one of the key responsibilities in the field of computer vision. It is indeed difficult to identify human in pictures because of several background effects such as occlusions [12], illuminated conditions and background clutters [13]. Previous techniques have been unsuccessful in real-world scenarios for detecting humans, as they took a longer period of time for detection and yielded outcomes that were not sufficiently accurate due to distance as well as changes in appearance [6]. Therefore, a universal representation of objects still continues to remain an open challenge in midst of such factors. Human detection is currently being utilized for many applications. Human detection is in the early stages in a number of use cases including pedestrian detection, e-health systems, abnormal behavior, person re-identification, driving assistance systems, crowd analysis, gender categorization, smart-video surveillance, human-pose estimation, human tracking, intelligent digital content management, and, finally, human-activity recognition [6,14–17].

The deep CNN is a dense computing framework in and of itself. With a large number of parameters and higher processing loads, followed by high memory access, energy consumption increases rapidly, thereby making it impossible to adopt the method for compact devices with minimal hardware resources. A feasible approach is a compressive, deep CNN technique for real-time applications and compact low-memory devices, which reduces the number of parameters, the cost of calculation, and power usage by compressing deep CNNs [18].

Over the past few years, the construction of tiny and effective network techniques to detect objects has become a point of discussion in the field of computer vision research. Acceleration and compression techniques are related primarily to the compact configuration of network architecture [19], knowledge distillation [20], network sparsity and pruning [21], and network quantization [22]. Various studies on network compression have advanced network models: for instance, SqueezeNet [23], which is a fire module based architecture; MobileNets [24], a depthwise separable filters based architecture; and, finally, the Shuf-

fleNet [25], a residual structure based network in which channel shuffle strategy and group pointwise convolution were incorporated.

Motivated by lightweight architectures, a novel compact model was proposed to detect humans for the portable devices that were absent in the current literature. Tiny-YOLO, which is the tiny version of the YOLO model, is used as the base architecture of this proposed model. YOLO is a faster and more accurate technique compared to other object detection models, and it has been enhanced since its first implementation, which includes v1-YOLO, v2-YOLO, and v3-YOLO. However, these architectures are not suitable for portable devices because of their large sizes and inability to maintain real-time performance in constrained environments. As mentioned, Tiny-YOLO is smaller than these models. However, it failed to achieve high accuracy, and speed remained unsatisfactory for low-memory devices.

This article proposes a model called ReSTiNet that is based on Tiny-YOLO. This model reduces the size of the model while simultaneously achieving higher accuracy and boosting detection speeds.The ultimate goal of this article is to develop a more capable human detection model for portable devices. Intelligent surveillance systems that use portable devices with less processing power can easily take advantages of this smaller and lighter model. This improves the performance and capabilities of the system without increasing the cost of the hardware or the amount of processing power it needs. Furthermore, lighter and faster models can be used in low-latency real-time human detection applications. The inspiration for ReSTiNet came from SqueezeNet, which use the fire module in order to decrease the total model parameter numbers and therefore compressed the overall size of the model. Determining the number of fire modules and where in the network they should be placed is one of the parts of integrating the fire module in Tiny-YOLO that presents one of the greatest challenges. The investigation of the residual connection between fire modules is still another key issue that needs to be addressed in order to improve detection accuracies and speeds even more. The useful feature of residual connections in Resnet [26] served as an inspiration for the implementation of residual connections within the fire modules of ReSTiNet. This was performed to ensure that the maximum amount of information flowed and to improve feature propagation throughout the architecture. In the end, dropout was used in ReSTiNet in order to circumvent the overfitting issue, attain an overall satisfactory level of performance, and lower the amount of computing effort required.

Prior to delving into the details of the study, it is essential to discuss the scope of the current effort. The following sections are the contents of this paper: Section 2 discusses the recent literature on human detection. In Section 3, the proposed ReSTiNet model for portable devices is explained. The experimental results are reported step-by-step in Section 4: system specification, dataset Specification, mAP, model training, ablation experiments of the proposed ReSTiNet, comparison with other lightweight models, and performance analysis of the proposed ReSTiNet. Finally, Section 6 concludes the article.

## 2. Related Literature: State-of-the-Art Methods

Human detection is the process of identifying each object in a static image or image sequences that are regarded to be human. Human detection is widely acknowledged to have advanced through two different historical periods in recent decades: "conventional human detection period (before 2012)" and "deep learning-based detection period (after 2012)", as illustrated in Figure 2.

Human detection is typically accomplished by extracting regions of interest (ROI) from an arbitrary image sample, illustrating the regions using descriptors, and then categorizing the regions as non-human or human, accompanied by post-processing processes [27].

In conventional techniques, human descriptors are generally designed by locally removing the features. A few examples include "edge-based shape features (e.g., [28])", "appearance features (e.g., color [29], texture [30])", "motion features (e.g., temporal differences [31])", "optical flows [32]", and their combinations [33]. Most of their functions are manually designed, which benefit from the ease of description and intuitively compre-

hending them. In addition, they were shown to perform well with limited collections of training datasets.

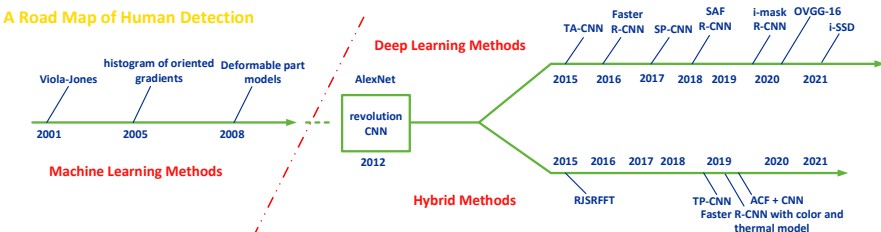

**Figure 2.** Human-detection milestones.

The Deformable Part-based Model (DPM) is the earlier state-of-the-art approach for detection process [34]. DPM is considered an extension of the histograms of the oriented gradients (HOG) model. The projected object is scored using the entire image's coarse global template as well as the six higher-resolution portions of the object. HOG is used to characterize every single input. Following this approach, HOG's multi-model can address the varying viewpoint problem. In the training phase, a latent support vector machine (latent SVM) was employed to decrease the detection drawback relative to the classification area. The coordinates of the component are considered as the latent element. This approach resulted in a massive impact due to its robustness.

Manually described features on the other hand, are unable to present more detailed information about the objects. In particular, they were challenged by the background, occlusion, motion blur, and illumination conditions. Hence, deep learning algorithms are regarded as relatively more efficient in human detection because they can learn more sophisticated features from images [35–37]. Although these initial deep algorithms have demonstrated some improvements over the classical models, these functions are still constructed manually, and the key concept is to expand the earlier models. Deep CNNs are also applied for the feature extraction in a few studies, for example [38]. A complexity perception cascade training for human detection was performed followed by the extraction of features.

Deep learning approaches are currently being used to address many identification problems in several ways. One of the most promising architectures is the Convolutional Neural Network (CNN). Deep CNNs can learn object features on their own; thus, they depend less on the object's classes. Training a class-independent method, contrastingly, means that more data will be used for learning as deep learning requires a significant volume of data relative to training a domain-specific method. Only a few articles have been published in the field of human detection using the CNNs method. Tian et al. [39] employed a CNN to learn human segmentation characteristics (e.g., hats and backpacks), but the network component leads to boosting the prediction accuracy by re-classifying the prediction item as negative or positive, rather than making predictions directly. Li et al. [40] included a sub-network relative to a novel network built on Fast R-CNN to deal with small-scale objects. Zhang et al. [41] straightforwardly examined a cross-class detection method (CNN), which involved faster R-CNNs performances on independent pedestrian detection, and came up with good findings. Among the three techniques, besides [39], which does not directly deal with detection, refs. [40,41] performed various experiments based on cross-class detection techniques. In [42], the authors suggested a system based on the combination of "Faster R-CNN" and "skip pooling" to deal with human detection issues. The architecture of "Faster R-CNN's region proposal network" is generalized to a multi-layer structure and finally combined with skip pooling. The skip pooling structure removes several interest regions from the lower layer and is fed to the higher layer, without considering the middle layer. In [43], the authors had suggested an enhanced mask R-CNN approach for real-time human detection that achieved 88% accuracy.

In [44], a deep convolutional neural network-based human detection technique was proposed using images that were used as input data to classify pedestrians and humans. The authors used the VGG-16 network as a backbone and the model had provided better accuracy on the "INRIA dataset". In [45], the authors combined a deep learning model with machine learning technique to achieve high accuracies with less computational time for human detection and tracking in real time. However, the model had a lower speed. In [46], the authors suggested a sparse network-based approach for removing irregular features and the developed approach was applied to a kernel-based architecture to reduce nonlinear resemblance across different features. This model, on the other hand, cannot be used for real-time detection and tracking.

H. Jeon et al. [47] resolved the human detection problem in extreme conditions by applying a deep learning-based triangle pattern integration approach. Triangular patterns are employed to derive more precise and reliable attributes from the local region. The extracted attributes are fed into a deep neural architecture, which uses them to detect humans in dense and occluded situation. In [48], K.N.Renu et al. proposed a deep learning-based brightness aware method to detect human in various illuminated conditions for both day and night scenarios. In [49], the authors cascaded aggregate channel features (ACF) with the deep convolutional neural network for quicker pedestrian and human detection. Then, a hybrid Gaussian asymmetric function was proposed to define the constraints of human perception. In [50], the authors proposed a single-shot multibox detector (SSD) to detect pedestrians. The SSD convolutional neural architecture extracts low features and then combines them with deep semantic information in the convolutional layer. Finally, humans are identified in still images. In the suggested technique, pre-selection boxes with different ratios are used, which increased the detection capability of the entire model.

In [51], the authors proposed a multi-stage cascade framework for coarse-to-fine human-object interaction (HOI) recognition understanding. The introduced method achieved first position in ICCV2019 Person Context Challenge (PIC-19) and also showed the excellent outcomes on V-COCO dataset. In [52], the authors developed a compressed, powerful, and effective architecture to resolve the instance-aware human part parsing issue. In the proposed method, structural information are used across a variety of human granularities, which makes the challenging task of person-partitioning easier.

## 3. The Proposed ReSTiNet for Low-Memory Devices

### 3.1. Motivation

The network structure of Tiny-YOLO is shown in Figure 3. This architecture consists of a total of nine convolutional layers followed by six max-pooling layers that are used to remove features of images along with one detection layer. This method uses convolutional layers containing 512 and 1024 filters that provide a large parameter density, large memory storage, and a lower detection speed. Another issue with Tiny-YOLO is its low detection accuracy. The network's irrational compression techniques may further decrease detection accuracies.

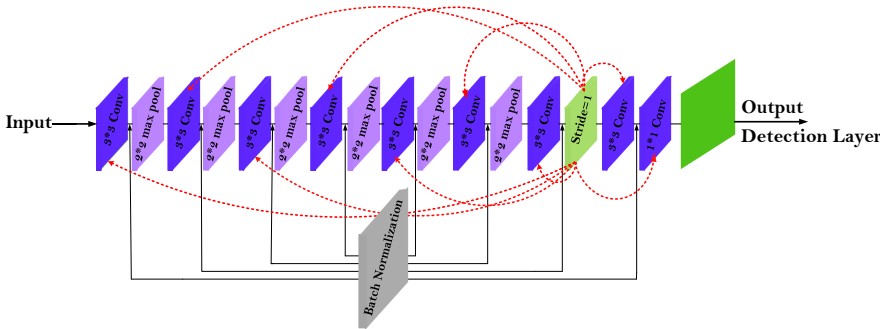

**Figure 3.** The structure of Tiny-YOLO.

Hence, in order to address such issues, ReSTiNet is introduced in this research, which directs towards the performance of the model's size as well as its accuracy. Algorithm 1 details the proposed ReSTiNet model.

---

**Algorithm 1** ReSTiNet pseudocode

---

　　**Input**: Input(shape= (input_size, input_size,3))
　　**Input**: learning_ rate, epoch, batch_size
　　**Input**: iou_ threshold, score_ threshold
　　**Output**: output_shape, mAP
　　**def** fire_module(model, fire_id, squeeze, expand)
　　**def** maxpooling (pool_size, stride)
　　**def** resnet_block (model, filters, reps, stride)
　　**def** mAP (model):
　　　　map = model.evaluate (generator, iou_threshold,
　　　　score_threshold, average_precisions)
　　　**return** map
　　**def** layer(conv, batchnorm, activation, maxpooling, dropout)
　　**def** main (){
　　　**create layer1**: ([16,3,1], norm_1, leakyReLU[.1], 2, null)
　　　**x** ← layer1
　　　　for i in range(2,3,4,5):
　　　　　create layer(i): ([32*(2**i), 3, 1], norm_ + str(i+2),
　　　　　leakyReLU[.1], 2, [0.20])
　　　　　**x** ← (x) (layer(i))
　　//return **x**
　　create fire_module1: (x, 2, 16, 64)
　　create fire_module2: (x, 3, 16, 64)
　　create maxpooling1: (3, 2)
　　create resnet_block1: (x, 64, 3, 1)
　　create fire_module3: (x, 4, 32, 128)
　　create fire_module4: (x, 5, 32, 128)
　　create maxpooling2: (3, 2)
　　create resnet_block1: (x, 128, 4, 2)
　　create fire_module5: (x, 6, 48, 192)
　　create fire_module6: (x, 7, 48, 192)
　　create fire_module7: (x, 8, 64, 256)
　　create fire_module8: (x, 9, 64, 256)
　　dropout ← 0.50
　　**return** mAP(x), output_shape(x)}

---

The goal of ReSTiNet is to develop a model that is smaller, swifter, and more capable at detecting humans on lightweight devices. The network's optimization is carried out by performing a reduction in parameters to an acceptable level rather than blindly decimating the convolution layers. SqueezeNet's fire module compresses the framework using a bottleneck network layer and widens the network module without significantly sacrificing detection accuracy. As a result, the introduction of fire module was carried out to achieve the performance of a faster as well as a smaller network structure. ReSTiNet then seeks achieve a higher accuracy in detection while simultaneously minimizing the parameters. Study [53] achieved a higher accuracy with a smaller number of parameters in which residual blocks were integrated between fire modules in the VGG-16 network. Thus, in between the fire modules lies the residual block, which is used in ReSTiNet to maximize the detection accuracy.

### 3.2. Construction of ReSTiNet

The structure of ReSTiNet is shown in Figure 4. The first five convolutional layers of Tiny-YOLO are retained in ReSTiNet. Layers with 512 and 1024 filters in the Tiny-YOLO are replaced with the fire modules, which shrink the model. Then, residual connections from Resnet-50 network inside the fire modules are integrated, which help the proposed model achieve a higher mAP. This article synthesizes three widely used approaches: Tiny-YOLO, ResNet, and the SqueezeNet method. The details of the implementations are as follows.

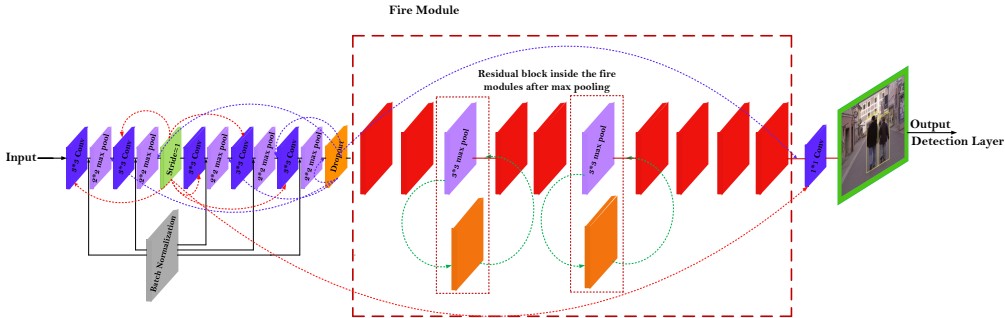

**Figure 4.** The structure of ReSTiNet. Fire modules are adopted from SqueezeNet, which shrinks the model. Then, residual connections are integrated from ResNet-50 network inside the fire modules to enhance the proposed ReSTiNet's efficiency.

### 3.2.1. Tiny-YOLO

A popular technique called "Tiny-YOLO", which is the smaller version of "You Only Look Once (YOLO)", was formulated to create a single step procedure that involved both the detection as well as the classification process. Upon a single appraisal of the input image, both the bounding box and class predictions are produced.

The distinguishing feature of this technique as opposed to the conventional models is that the class as well as bounding box predictions are performed at the same time. The procedure is as follows: Firstly, the image that is considered as the input is split across the $S \times S$ grid. Secondly, every single grid cell is assigned with a confidence score, which contains the respective bounding box. The probability or chances that the object is present in every bounding box is referred to as the confidence score and is mathematically given by the following:

$$C = \Pr(Object) * IOU_{pred}^{truth} \tag{1}$$

where term IOU ("intersection over union") is defined to be a fraction that numerically lies within the limits of [0, 1]. The overlapped area in between the ground truth as well as the bounding box predictor is termed as the intersection. The entire region between the ground truth and the predictor is known as the union. In ideal terms, the IOU must be closer to 1, which implies that the ground truth is approximately equal to the bounding-box predictor.

Similarly, the conditional class probability C is also predicted by individual grid cells while the bounding boxes are created. Thus, for every cell, the class-specific probability function is expressed as follows.

$$\begin{aligned} \Pr(Class_i|Object) * \Pr(Object) * IOU_{pred}^{truth} \\ = \Pr(Class_i) * IOU_{pred}^{truth}. \end{aligned} \tag{2}$$

### 3.2.2. Fire Module of ReSTiNet

The introduction of the fire module under ReSTiNet was to decrease the number of parameters as well as escalate the width and depth of the entire network. This was performed in order to ensure the accuracy of detection. This model consists of both expand as well as the squeeze components so that the model's network tends to expand and

compress. The compress or squeeze component utilizes the convolutional layer with a size of $1 \times 1$ introduced by NIN as a substitute for the usual layer with the size $3 \times 3$. In order to decrease the number of parameters, the model that follows the $1 \times 1$ technique was found to be more efficient. Additionally, the accuracy of detection does not reduce significantly as the training parameter is only a single variable that should be learnt. During the expansion, both the models with sizes $1 \times 1$ as well as $3 \times 3$ are typically used. Finally, the arrived outputs from the respective convolutional layers are concatenated at the concatenation layer.

For a convolutional layer, the parameters are given as $c_i$, the number of channel input variables, $k$ as the kernel size, and $c_o$ as the number of channel output variables. Using Equation (3), the value of the number of parameters for the convolutional layer is then calculated. The number of channel inputs is $c_i$ for the fire module; $k_{s_1}$ is the kernel size of the squeeze component, and $s_1$ is the number of channel output variables. If the value of $k_{s_1}$ is assigned to 1, a reduction in a large number of model parameters for the squeeze component is possible. The number of channel input variables is $s_1$ followed by the kernel sizes $k_{e_1}$ and $k_{e_3}$ for the expanded component. The total number of channel output variables is the sum of $e_1$ and $e_3$. Using Equation (4), the number of model parameters is calculated. Figure 5 illustrates the structure of the fire modules in ReSTiNet.

$$P_{conv} = (c_i \times k^2 + 1) \times c_o \tag{3}$$

$$
\begin{aligned}
P_{fire} = (c_i \times k_{s_1}^2 + 1) \times s_1 + (s_1 \times k_{e_1}^2 + 1) \\
\times e_1 + (s_1 \times k_{e_3}^2 + 1) \times e_3
\end{aligned}
\tag{4}
$$

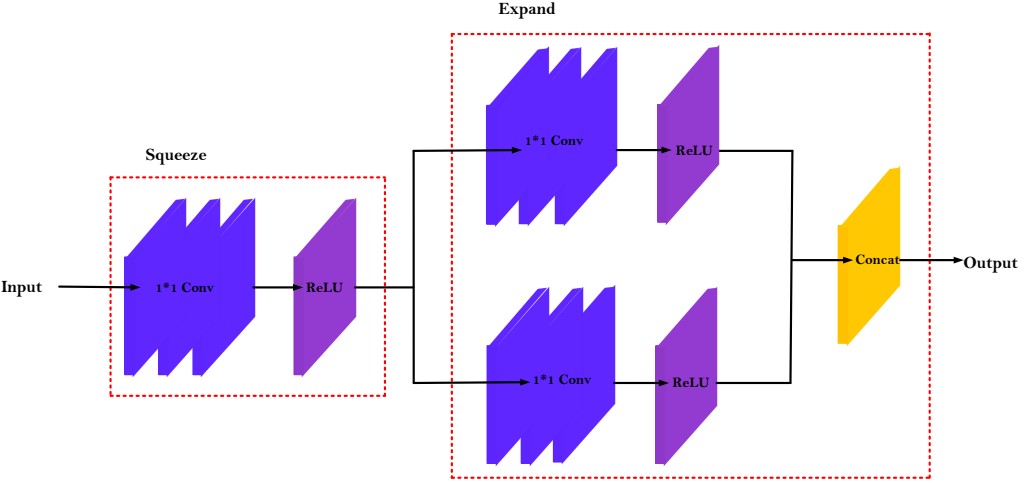

**Figure 5.** The structure of Fire Module. It is made up of two layers: squeeze and expand. The squeeze layer consists of a small number $1 \times 1$ filters, and the expand layer consists of a small number of $3 \times 3$ and $1 \times 1$ filters.

The ability with which the fire can be utilized more effectively depends on the appropriateness of the position of the fire module within the network. ReSTiNet architecture comprises a total of eight fire modules. In the ReSTiNet network, the sixth layer is replaced with the initial four fire modules where the former contains 512 filters followed by down-sampling technique. Layers seventh and eighth containing 1024 filters are replaced with four other fire modules, and this is carried out before the $1 \times 1$ convolutional layer and detection layer. However, the choice of the number of channel inputs $c_i$ is not bounded, while choosing a large number of channel inputs would lead to reduction in parameters.

### 3.2.3. Residual Block between Fire Modules

The optimization trajectory will follow a negative slope due to degradation when it is expected of depth to provide an enhanced detection accuracy. Relative to the conclusions

derived from other neural networks, it is observed that the error is typically higher in deep CNN architectures [53]. In study [26], the authors developed a degradation resolution that enables a subset of stacked layers to accept the existing residual mapping. This is the area where the degradation typically halts the layers in order to be congruent with the standard subsidiary mapping. Formula (6) represents the subsidiary mapping rather than Formula (5), where $H(x)$ is the desired mapping, and $F(x)$ is the learned residual mapping. The actual mapping is modified into $F(x) + x$. In study [26], the authors found that optimization is relatively easier in a residual-based mapping than the primary one.

$$F(x) = H(x) \tag{5}$$

$$F(x) := H(x) - x \tag{6}$$

$$H(x) = F(x) + x \tag{7}$$

However, one or more layers were ignored during "shortcut connections", as mentioned in studies [26,53]. "Shortcut connections" are expressed in Equation (7) [26]. Study [26] utilized "shortcut connections" in order to conduct identity mappings. The fusing of stacked layer outputs is performed with that of the "shortcut connections" output values. The latter possesses the advantage of being parameter-free, thereby using minor values during the computational process. Paper [54] developed highway-based networks by merging "shortcut connections" and "gating functions" along with their parameters. The possibility of optimization using a "stochastic gradient descent (SGD)" is another benefit of "shortcut connections" [26]. It is easier to integrate "identity shortcut connections" using deep learning open-source libraries [26,53].

We integrated "residual learning" from ResNet-50 within ReSTiNet architecture followed by a down-sampling technique after the 2nd and 4th fire modules. The building residual block is expressed in Equation (8).

$$y = F(x, W_i) + x \tag{8}$$

Terms $y$ and $x$ represent the output and input vectors of the layers, respectively. The mentioned function, $F(x, W_i)$, is nothing but the residual mapping to be learnt. As in Figure 2, there are 2 layers, $F = W_2\sigma(W_1 x)$, in which the term $\sigma$ represents the ReLU functionl; for reducing the complexity of notations, many biases were appropriately removed. The process, $F + x$, was carried out by the use of a "shortcut connection" and "elemental-wise addition" operation upon which the second ReLU (non-linearity) function was made use of. The "shortcut connections" in Equation (8) add neither more parameters nor complexity to the computation [26].

### 3.2.4. Dropout in ReSTiNet

The mask that neutralizes the effects caused by neurons in the succeeding layer is termed as the dropout layer. This mask tends to stabilize the neurons and keeps the others unchanged. This layer is important while training CNNs since they counteract the effects of overfitting on the data that needs to be trained. Otherwise, an influence from the initial batch of samples will be present on the learning and causes disproportionate results in the performance. Thus, the efficiency in learning the features will be deeply affected; it further delays the arrival of such results in later batches [55]. The common practice of a dropout is to use a small value within the range of 20–50% of neurons, with 20% being a decent starting point. A probability that is too low has no impact, whereas a value that is too large results in the network's under-learning [56]. In the convolutional layers (2nd–5th), 0.2 and, after the fire module, 0.5 dropouts are used in the proposed ReSTiNet network to overcome the overfitting problem.

### 3.2.5. Loss Function of ReSTiNet

The custom loss function is utilized in this study, unlike Tiny-YOLO, which consists of three parts: error in prediction coordinate, error in IOU, and classification error.

The error in coordinate prediction is described as follows:

$$
\begin{aligned}
Error_{coord} = \\
\lambda_{coord} \sum_{i=0}^{s^2} \sum_{j=0}^{B} \mathbf{L}_{ij}^{obj} \left[ (x_i - \hat{x}_i)^2 + (y_i - \hat{y}_i)^2 \right] \\
+ \lambda_{coord} \sum_{i=0}^{s^2} \sum_{j=0}^{B} \mathbf{L}_{ij}^{obj} [(\sqrt{w_i} - \sqrt{\hat{w}_i})^2 + \left( \sqrt{h_i} - \sqrt{\hat{h}_i} \right)^2]
\end{aligned}
\tag{9}
$$

where $s^2$ denotes the grid cell number of all scale. $B$ represents the bounding-box number for every grid. $L_{ij}^{obj}$ defines target of the $i$-th grid cell, which falls in the $j$-th bounding box. $(\hat{x}_i, \hat{y}_i, \hat{w}_i, \hat{h}_i)$ and $(x_i, y_i, w_i, h_i)$ represent the center coordinate, height, and width of the predicted box and the ground truth, respectively.

The IOU error is described as follows:

$$
\begin{aligned}
Error_{IOU} = \sum_{i=0}^{s^2} \sum_{j=0}^{B} \mathbf{L}_{ij}^{obj} \left( C_i - \hat{C}_i \right)^2 \\
+ \lambda_{noobj} \sum_{i=0}^{s^2} \sum_{j=0}^{B} \mathbf{L}_{ij}^{noobj} \left( C_i - \hat{C}_i \right)^2
\end{aligned}
\tag{10}
$$

where $\hat{C}_i$ and $C_i$ define the predicted and true confidence, correspondingly.

The classification error is defined as follows:

$$
Error_{cls} = \sum_{i=0}^{s^2} \mathbf{L}_i^{obj} \sum_{c \in classes} (p_i(c) - \hat{p}(c))^2.
\tag{11}
$$

where $\hat{p}_i(c)$ denotes the predicted value, while $p_i(c)$ denotes the target's true probability. From the above, the final loss function is shown in Equation (12).

$$
\begin{aligned}
Loss = Error_{coord} + Error_{IOU} + Error_{cls} \\
= \lambda_{coord} \sum_{i=0}^{s^2} \sum_{j=0}^{B} \mathbf{L}_{ij}^{obj} \left[ (x_i - \hat{x}_i)^2 + (y_i - \hat{y}_i)^2 \right] \\
+ \lambda_{coord} \sum_{i=0}^{s^2} \sum_{j=0}^{B} \mathbf{L}_{ij}^{obj} [(\sqrt{w_i} - \sqrt{\hat{w}_i})^2 + \left( \sqrt{h_i} - \sqrt{\hat{h}_i} \right)^2] \\
+ \sum_{i=0}^{s^2} \sum_{j=0}^{B} \mathbf{L}_{ij}^{obj} \left( C_i - \hat{C}_i \right)^2 \\
+ \lambda_{noobj} \sum_{i=0}^{s^2} \sum_{j=0}^{B} \mathbf{L}_{ij}^{noobj} \left( C_i - \hat{C}_i \right)^2 \\
+ \sum_{i=0}^{s^2} \mathbf{L}_i^{obj} \sum_{c \in classes} (p_i(c) - \hat{p}(c))^2.
\end{aligned}
\tag{12}
$$

### 3.3. Time Complexity, Success, and Challenge of ReSTiNet

In this section, time complexities of the proposed ReSTiNet with its success and challenge are described.

### 3.3.1. Time Complexity

In the proposed algorithm, some operations occur only once and their time complexity is $O(1)$. However, in ReSTiNet, different methods have iteration, and their time complexity is $O(n^2)$. Therefore, the time complexity of our proposed algorithm is as follows: $O(1) + O(n^2) = O(n^2)$. Therefore, we define this algorithm as having a Quadratic Time

Complexity to indicate that as the size of the input increases, the amount of time needed to run it increases accordingly. Informally, Quadratic Time Complexity represents an algorithm for which its performance is directly proportional to the squared size of the input data set.

### 3.3.2. Advantage of the Model

This proposed method is easily adaptable; therefore, this process can be applied to compress various current deep CNN models. As human detection is the first phase of many applications, this developed method can be used for pedestrian detection and pose estimation with low-memory devices.

### 3.3.3. Challenge of the Model

ReSTiNet employs fire modules that reduce the model's parameter and, thus, the computational cost. However, the procedure still requires a significant amount of processing to be performed on a portable device. As a result, the architecture is still trained on a machine (i.e., remote server) capable of handling this computationally intensive method.

## 4. Experimental Results

Initially, the experimental environment setups, datasets, and evaluation criteria (mAP) are described in this segment of the article. The performance is then compared based on training time, mAP, and model size metrics. Moreover, to validate the advantage of ReSTiNet performance over alternative lightweight networks, we conducted comprehensive experiments to verify the findings of the performance comparison.

### 4.1. System Specification

The Tesla K80 is used to train the ReSTiNet model and also to evaluate the detection speed of the architecture. The Tesla K80 is a pro graphics card launched by NVIDIA. Tesla K80 is built on the GK210 GPU and manufactured using 28 nm technology. The GK210 GPU has a 561 mm² die area with 7100 million transistors. The Tesla K80 integrates two GPUs to boost the performance. The configuration of the Tesla K80 is provided in Table 1.

**Table 1.** Configuration of Tesla K80.

| Computing Platform | Graphics Processor | Memory |
|---|---|---|
| Tesla K80 | GK210 $\times$ 2, 2496 $\times$ 2 shading units, 208 $\times$ 2 TUMs, 48 $\times$ 2 ROPs | 12 GB $\times$ 2, 384 bit $\times$ 2, GDDR5, 240.6 GB/s $\times$ 2 |

Ubuntu-16.04 LTS is used as base operating system with 62 GB RAM, NVIDIA CUDA v10.2, NVIDIA cuDNN v7.6.5. The script is written in python v2.7 with TensorFlow v1.14.0, Keras v2.2.2, cv2, NumPy v1.16.4.

### 4.2. Data-Set Specification

This study makes use of the "MS COCO" [57] and "Pascal VOC" [2] datasets. Generally, object detection, image classification, and segmentation are performed with these two datasets. The "Pascal VOC" dataset consists of "Pascal VOC 2007" and "Pascal VOC 2012". There are 8540 images of human beings from the "Pascal VOC" dataset used for this experiment. "MS COCO" is more challenging while "Pascal VOC" is easier to train. Generally, the performance on the MS COCO dataset of a method for object detection models is more inclined. There are 45,174 images of human beings used from the "MS COCO-train2014" dataset for accomplishing this study. Both datasets are split 80/20 for training and validation, respectively. The IOU ("intersection over union") is set 0.5 by default for both datasets while calculating mAP values. The "INRIA" dataset [6] (1208 images) is used to test the proposed ReSTiNet model's detection speed.

### 4.3. Evaluation Criteria (mAP)

The mean average precision (mAP) metric is utilized to estimate the performance of the introduced ReSTiNet and the baseline architectures. The mAP scores are reported for both "MS COCO" and "Pascal VOC".

Average Precision (AP): The recall/precision curve is used to assess the output performance for a specific class and task. Precision is the ratio between the relevant and retrieved examples explained in Equation (13).

$$precision = \frac{|\{relevant\ instances\} \cap \{retrieved\ instances\}|}{|\{retrieved\ instances\}|} \tag{13}$$

The rate of recall is described as the ratio of the total number of relevant examples to the total positive instances. The average precision is utilized for evaluating precision over multiple equidistant recall levels:

$$AP = \sum_{k=0}^{k=n-1} [Recalls(k) - Recalls(k+1)] * Precisions(k) \tag{14}$$

where $n$ defines the number of the threshold.

The mAP ("mean average precision") is employed to calculate the C class's average precision:

$$mAP = \frac{1}{C} \sum_{i \in \{0,1,2,...,C\}} AP(c_i) \tag{15}$$

where $AP(c_i)$ defines the average precision for the class of $c_i$.

### 4.4. Model Training

The pre-trained weight daraknet19.conv model is imported into ReSTiNet before the training started on both "MS COCO" and "Pascal VOC" datasets. ReSTiNet takes 416*416 as the size of the input. The learning rate of ReSTiNet is 0.001, and the batch size is 16 with 50 epochs. MS COCO has a max iteration batch number of 504K, whereas Pascal VOC has a max iteration batch number of 129 K. Table 2 represents the model's trained hyperparameters.

**Table 2.** Hyperparameters used in the ReSTiNet.

| Hyperparameter | Range |
|---|---|
| input size | $416 \times 416$ |
| learning rate | 0.001 |
| activation | Leaky ReLU ($\alpha = 0.1$), ReLU |
| batchsize | 16 |
| no. of epoch | 50 |
| optimizer | adam ($\beta_1 = 0.9$, $\beta_2 = 0.999$, $\epsilon = 1 \times 10^{-8}$) |
| loss function | custom loss |
| dropout | 0.2, 0.5 |
| iou_threshold | 0.5 |
| score_threshold | 0.5 |

### 4.5. Ablation Experiments of the Proposed ReSTiNet

We have conducted ablation experiments in the ReSTiNet network by sequentially adding fire modules, residual connections, and dropout layers to demonstrate the impact of these methods on ReSTiNet's performance. Table 3 shows the results (mAP) of the proposed model, ReSTiNet, and the original Tiny-YOLO on the "Pascal VOC" and the "MS COCO" dataset.

**Table 3.** Ablation experiments: Tiny-YOLO vs. ReSTiNet.

|  | | Tiny-YOLO | | ReSTiNet |
|---|---|---|---|---|
| fire module | |  | ✓ | ✓ |
| residual learning | |  |  | ✓ |
| dropout | |  | ✓ | ✓ |
| MS COCO mAP(%) | | 19.0 | 24.37 | 27.31 |
| Pascal VOC mAP(%) | | 42.21 | 55.67 | 63.79 |

Detection accuracy is commonly evaluated using the mAP. The new proposed model achieved 27.31% mAP on the MS COCO dataset, whereas Tiny-YOLO obtained 19% mAP. The dropout layer and residual connections helps the ReSTiNet model in achieving higher accuracies than Tiny-YOLO. ReSTiNet achieves 63.79% mAP on the Pascal VOC dataset; on the other hand, Tiny-YOLO reaches 42.21% mAP. The use of residual connections between the fire modules and the dropout layer significantly contributes to the increase in mAP without requiring an excessive number of parameters. Utilizing residual connections and dropout improves mAP by 12.06% on "MS COCO" and 14.59% on "Pascal VOC" based on adding fire modules. In addition, employing the dropout layer helps reduce the training time and helps the model from the over-fitting problem. ReSTiNet outperforms Tiny-YOLO, showing 43.74% and 51.09% improvements on the "MS COCO" and "Pascal VOC" datasets, respectively.

Detection Time, Parameter, and FLOPs Comparison between Tiny-YOLO and ReSTiNet

The entire testing time is calculated for 1208 images from the "INRIA Person" dataset using the Tesla K80. Table 4 shows the average test time, total parameter, and FLOPs for both Tiny-YOLO and ReSTiNet models. As observed, the overall time needed to detect 1208 images using the ReSTiNet is less than 40 s. ReSTiNet outperforms Tiny-YOLO in terms of detection speeds. Tiny-YOLO completes the detection in more than 74 s. When compared to Tiny-YOLO, the detection speed of ReSTiNet is improved by 49.2%. On the other hand, it has been observed that ReSTiNet has 80.90% less parameters than Tiny-YOLO, and the FLOP's amount is also reduced by 34.47%.

**Table 4.** Detection time, parameter, and FLOP comparison.

|  | Tiny-YOLO | ReSTiNet | Dataset |
|---|---|---|---|
| Avg. test time | 74.486 (s) | 37.514 (s) | INRIA |
| Model parameters | 11.043 (m) | 2.109 (m) | - |
| FLOPs | 11.552 (bn) | 7.570 (bn) | - |

*4.6. ReSTiNet Performance Comparison with Other Lightweight Methods*

ReSTiNet is compared with the other lightweight state-of the-art networks, such as MobileNet, SqueezeNet, and Tiny-YOLO in order to analyze the proposed model's further improvement. The "Pascal VOC" customized dataset is used to train the MobileNet and SqueezeNet models. The training operation is performed on the Tesla k80, which operates in similar experimental settings as ReSTiNet.

The comparative findings of the four models are summarized in Table 5. As shown in Table 5, ReSTiNet outperforms Tiny-YOLO and MobileNet in terms of model size and achieves higher mAP compared with all three models. The model size of SqueezeNet is very impressive, while resulting in very low mAP. The proposed model is 10.7 MB, which is larger than SqueezeNet yet smaller than MobileNet and TinyYOLO. Compared with Tiny-YOLO, ReSTiNet reduces the model size of 82.31%, which is suitable for portable devices. ReSTiNet shows 51.09%, 35.38%, and 53.67% improvements in terms of mAP on Tiny-YOLO, MobileNet, and SqueezeNet, respectively.

**Table 5.** ReSTiNet vs. other lightweight models.

| Network | mAP (%) | Model Size (MB) |
|---------|---------|-----------------|
| MobileNet | 47.12 | 13.5 |
| SqueezeNet | 41.51 | 3.0 |
| Tiny-YOLO | 42.21 | 60.50 |
| ReSTiNet | **63.79** | **10.7** |

## 5. Performance Analysis of ReSTiNet

Tiny-YOLO is used as the backbone architecture for the proposed ReSTiNet model. As Tiny-YOLO has several layers with 512 and 1024 filters, it has a large number of parameters, its speed is slow, and its model size is large. A replacement is carried out, thereby using the fire module instead of the sixth, seventh, and eighth layer present in the Tiny-YOLO method as the fire module contains far lower numbers of parameters as opposed to its counterpart filter of size $3 \times 3$.

The input channels also decreased to the filters with a size of $3 \times 3$. Then, by multiplying the number of filters as well as the input channel values, the net parameters present in the fire module can be calculated. By reducing the input channel and filter count, a deep CNN network can be designed containing only fewer number of parameters. Table 6 shows that the parameters that are reduced in the layer containing 256 filters are numerically lower relative to the layer with 512 filters.

**Table 6.** Parameter numbers comparison between fire modules and convolutional layers.

| Conv. Layer | Input Channel | Output Channel | Kernel Size | Conv. Layer (Parameters) | Fire Module (Parameters) |
|-------------|---------------|----------------|-------------|--------------------------|--------------------------|
| 1 | 3 | 16 | 3 | 448 | 184 |
| 2 | 16 | 32 | 3 | 4680 | 740 |
| 3 | 32 | 64 | 3 | 18,496 | 2888 |
| 4 | 64 | 128 | 3 | 73,856 | 11,408 |
| 5 | 128 | 256 | 3 | 295,168 | 45,344 |
| 6 | 256 | 512 | 3 | 1,180,160 | 180,800 |
| 7 | 512 | 1024 | 3 | 4,719,616 | 722,048 |
| 8 | 1024 | 512 | 3 | 4,719,616 | 722,048 |

If the goal is to further decrease the number of model parameters, a replacement with a larger number of channel inputs in the convolutional layer is required while distributing such layers in the middle and the end components of the ReSTiNet module.

It was found that the accuracy of detection becomes poor if fire modules substitute the total convolutional layers since fire modules replace certain convolutional layers with a limited number of filters. If the convolutional layers (first five) with a fewer number (less than 256) of convolution filters are retained instead of being substituted by fire modules, the rate of accuracy can improve by 6.2 percent and the size of model can increase by 1.6 megabyte. Thus, ReSTiNet retains the frontal (first five) convolutional layers while replacing the convolutional layers (three) with eight fire modules at the end of Tiny-YOLO.

A simplistic method to compress the network is to decrease the number of layers in the network, network scaling factor, and to utilize networks that are considered shallow. However, the degree of freedom to efficiently compress such networks is limited and more distant from existing DNN models [58]. Ba et al. [59] suggested a training procedure of shallow neural networks that best simulates the deep models, but there has been an increase in the number of parameters. In study [60], the authors had shown that the degree of expansion possesses the capability to be exponentially grown as a function of increasing depth. However, networks that are too shallow do not play the role of substitution for deeper networks. As illustrated in Figure 4, there are five pooling layers after the five convolutional layers. It contains a total of eight fire modules with a depth value of 2 followed by convolutional layer with a kernel size $1 \times 1$ in the ReSTiNets architecture.

The above mentioned eight fire modules in ReSTiNets replaced the three convolutional layers from the last layer in Tiny-YOLO. As a result, the net depth attains a value of 29, which is exactly twelve layers deeper in physical depth compared to Tiny-YOLO thereby raising the network's accuracy.

All max-poolings are set to $3 \times 3$ in size followed by the down-sampling technique later within the architecture. This in turn yields several layers with large activation maps [61]. Such layers provide activation maps with a minimum of $1 \times 1$ spatial resolution and typically in higher orders at other times.

Activation maps' width and height can be determined using a set of variables, namely the input data size and various choices of layers in which down-sampling more likely tends to occur. The down-sampling strategy has been accomplished in studies [62–64] using a stride that is larger than one during a choice of convolutional or pooling layers. It was concluded that a large number of layers contain smaller activation maps when the initial layers are set to larger stride parameters. The authors in [65] detected improved classification accuracies after implementing down-sampling strategies into four distinct CNN networks [53].

Then, residual connections are integrated to examine whether it can increase the efficacy of the Tiny-YOLO network while making the model quicker and smaller at the same time. The concept of the fire module [23] is modified by adding residual connections at strategic locations across the network. The model does not experience an increase in complexity apart from a bit of computation associated with the collection operation as the residual connections do not have any parameters. This model employs the dropout layer to handle the over-fitting issue and speed up data processing. The dropout method disregards the randomly chosen neurons during the training period.

Figures 6 and 7 show the detection results for both proposed ReSTiNet and Tiny-YOLO models. From all figures, it can be seen that the proposed model detects human objects with a higher accuracy, while Tiny-YOLO can sometimes miss objects and recognize non-human objects as human. These scenarios are shown in Figures 6c and 7b . The proposed ReSTiNet sometimes misses people in a dense scenario shown in Figure 7a. In this scenario, there are five people on the wall. Of these five people, ReSTiNet detected only four. However, the detection rate is still better than that of Tiny-YOLO, which detected only two out of the five people, but the proposed method can still be improved. The images showing the results of the detection are available in full size at the following URLs: Figure 6: https://i.ibb.co/6FhDYf5/P1-comp.png (accessed on 4 September 2022); Figure 7: https://i.ibb.co/tDs6xPB/P2-comp.png (accessed on 4 September 2022).

The ReSTiNet architecture that has been suggested has a significantly reduced number of parameters while also preserving a greater amount of information flow throughout the model. It detects humans more quickly than other lightweight models, and its performance in terms of detection time and mAP score is superior to that of those models. This is despite the fact that the model itself is quite compact.

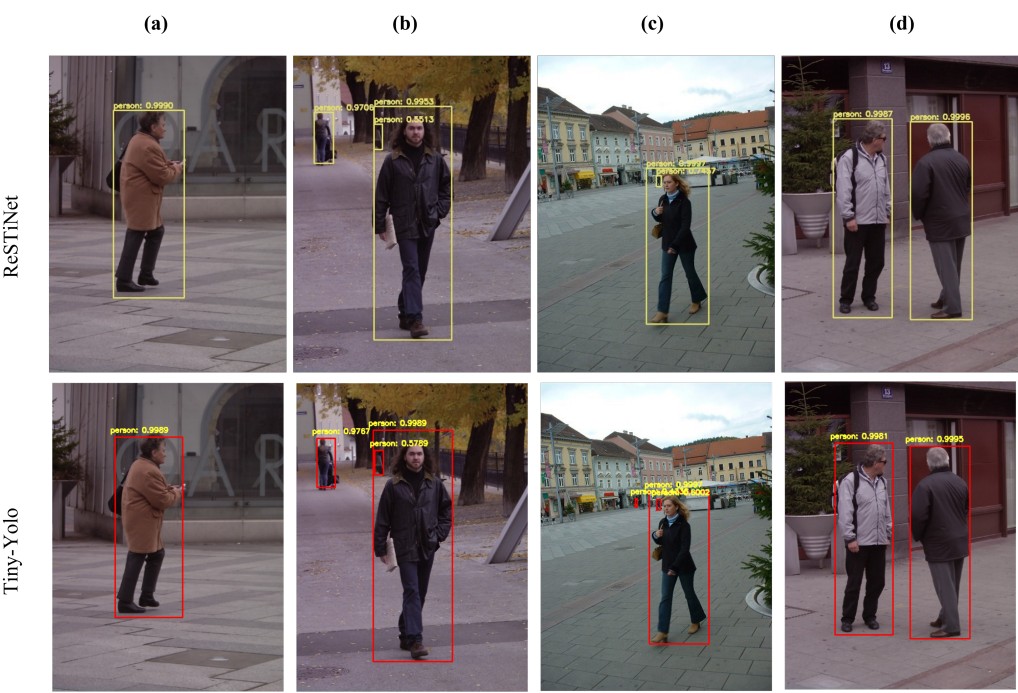

**Figure 6.** Detection results with confidence values for the proposed ReSTiNet and Tiny-YOLO model in a sparse scenario. (**a**–**d**) presents comparison for four images for the models.

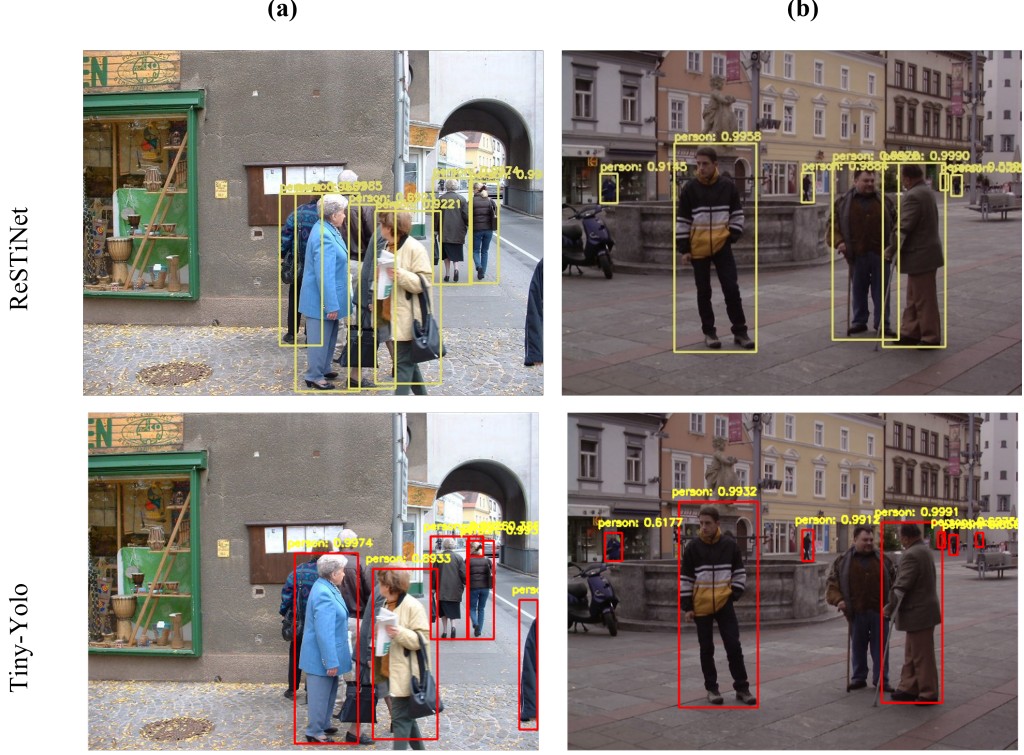

**Figure 7.** Detection results with confidence values for proposed ReSTiNet and Tiny-YOLO model in a dense scenario. (**a**,**b**) presents comparison for two images for the models.

## 6. Conclusions

In this article, ReSTiNet, a compact human-detection method, is proposed for portable devices, and it focuses on issues related to size, speed, and accuracy. The suggested method reduces the size of the previously popular Tiny-YOLO algorithm while improving the following characteristics: improving detection performance, reducing model size,

resolving overfitting issues, and outperforming existing lightweight models in terms of mAP. ReSTiNet is constructed by first incorporating the fire modules from SqueezeNet inside the Tiny-YOLO with the aim of minimizing the model's size. Following that, the fire module numbers and their placement have been investigated in the model's architecture. The residual connection inside the fire modules in Tiny-YOLO is integrated from the Resnet model. The residual connection helps maximize feature propagation and information flow within the network, with the aim of further improving the developed ReSTiNet's detection speed and accuracy. Using the dropout layer in the convolutional layer and at the end of the fire module helps resolve the overfitting problem in ReSTiNet. The experimental results show that ReSTiNet outperforms Tiny-YOLO in terms of efficiency. ReSTiNet also exhibits comparable performances when compared to lightweight models such as MobileNet and SqueezeNet with respect to the model's size and mAP. The findings show the effectiveness of ReSTiNet for portable devices. The developed algorithm can be simply modified and completely incorporated into a variety of different deep convolutional neural networks for compression. The performance of ReSTiNet will be further optimized in future for high-resolution images, particularly for the EuroCity Persons dataset.

**Author Contributions:** Conceptualization, S.S.S., D.R.A.R. and S.M.; methodology, S.S.S., D.R.A.R. and S.M.; software, S.S.S., M.M.E. and M.S.U.M.; validation, D.R.A.R., S.M., M.M.E. and M.S.U.M.; investigation, S.S.S., D.R.A.R., S.M., M.M.E. and M.S.U.M.; writing—original draft preparation, S.S.S.; writing—review and editing, S.S.S., D.R.A.R., S.M., M.M.E. and M.S.U.M.; supervision, D.R.A.R. and S.M.; funding acquisition, D.R.A.R. All authors have read and agreed to the published version of the manuscript.

**Funding:** YUTP-FRG (Cost Centre 015LCO-242), Universiti Teknologi PETRONAS (UTP), funded this work.

**Institutional Review Board Statement:** Not applicable.

**Informed Consent Statement:** Not applicable.

**Data Availability Statement:** Not applicable.

**Acknowledgments:** The authors would like to extend their gratitude to the High Performance Computing Centre (HPCC), Department of Computer and Information Sciences, Universiti Teknologi PETRONAS, Malaysia, for providing opportunities in the use of resources.

**Conflicts of Interest:** The authors declare no conflict of interest.

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
