# Peer review of "ReSTiNet: On Improving the Performance of Tiny-YOLO-Based CNN Architecture for Applications in Human Detection"

_applsci, doi:10.3390/app12189331_

Round 1

Reviewer 1 Report

Pros:

The paper is overall well written and easy to understand.

The results look promising

Cons:

Fig.6 is not informative, which is identical to the one in ResNet paper. Please consider dropping it.

It is unclear to me the meaning of the title of Sec. 3.3.2.

Please provide a comparison of ReSTiNet with Tiny-YOLO in terms of model parameters, FLOPs, or inference speed. 

The related work is not comprehensive. Some relevant human-centric analysis methods should be discussed, e.g., Cascaded human-object interaction recognition and Differentiable multi-granularity human representation learning for instance-aware human semantic parsing.

Is 416x416 used for all the datasets? Why is this size selected? How will different input sizes affect the performance?

Author Response

Point 1: The paper is overall well written and easy to understand.

Response 1: The authors are thankful to the reviewer for the positive comments.

Point 2: The results look promising

Response 2: The authors are thankful to the reviewer for the positive comments.

Point 3: Fig.6 is not informative, which is identical to the one in ResNet paper. Please consider dropping it.

Response 3: The authors would like to thank the reviewer for the feedback. As per the reviewer’s feedback, we have updated the manuscript. Updates are available in line no. 353-364.

Point 4: It is unclear to me the meaning of the title of Sec. 3.3.2.

Response 4: The authors would like to thank the reviewer for the concern. The section name has been modified in the updated manuscript. Please see the line no. 401.

Point 5: Please provide a comparison of ReSTiNet with Tiny-YOLO in terms of model parameters, FLOPs, or inference speed.  

Response 5: The authors would like to thank the reviewer for the concern. To address the reviewer’s concern, model parameters and FLOPs comparison are added in the updated manuscript. The changes can be found at section- 4.5.1, line no. 477-486.

Point 6: The related work is not comprehensive. Some relevant human-centric analysis methods should be discussed, e.g., Cascaded human-object interaction recognition and Differentiable multi-granularity human representation learning for instance-aware human semantic parsing.

Response 6: The authors thank the reviewer for providing good direction for the improvement of the manuscript. The related work section has been updated as per the direction of the reviewer. Please see the updated manuscript, line no. 243-249.

Point 7: Is 416x416 used for all the datasets? Why is this size selected? How will different input sizes affect the performance?

Response 7: The author thanks the reviewer for his concern. We compared the performance of our model with that of Tiny Yolo, which has an input size of 416*416 for its baseline models. To compare our model with Tiny Yolo's model, the image size is kept constant for both models. The authors would like to thank the reviewer again for his question about the performance at different input sizes, which leads to another research experiment that can be conducted in the future as a continuation of this experiment.

Reviewer 2 Report

This paper proposes new CNN architecture to improve the performance of the baseline model.

However, to convince readers about its capability and adaptability, the author should:

  1. Add more baseline architecture, not only Tiny-YOLO
  2. Show some qualitative results to prove the proposed method's improvement

In addition,

  1. The importance of the proposed method in improving human detection is not highlighted
  2. Discussion related to the human detection task is not significant 

Author Response

Point 1: This paper proposes new CNN architecture to improve the performance of the baseline model.

Response 1: The authors are thankful to the reviewer for the positive comments.

Point 2: Add more baseline architecture, not only Tiny-YOLO

Response 2: The authors thank the reviewer for providing good direction for the improvement of the manuscript. The changes can be found at the section 4.6. (ReSTiNet performance comparison with other lightweight methods), line no. 487-500.

Point 3: Show some qualitative results to prove the proposed method's improvement

Response 3: The authors would like to thank the reviewer for his original concern. In response to the reviewer's concern, some results have been added to Section 5: Performance Analysis of ReSTiNet in lines: 556-566. The added qualitative result shows the improvement of the proposed model over the tiny-yolo model.

Point 4: The importance of the proposed method in improving human detection is not highlighted

Response 4: The authors would like to thank the reviewer for his concerns, which will improve the quality of the manuscript. As suggested by the reviewer, the importance is discussed in Section 1: Introduction, line numbers: 127-155.

Point 5: Discussion related to the human detection task is not significant

Response 5: The authors would like to thank the reviewer for the suggestion. As per the reviewer’s suggestion, the discussion is updated by providing some qualitative results. Please see lines 487-500.

Reviewer 3 Report

1. The abstract must be clear: what dataset was used to test the ReSTiNet, the new proposed architecture (probably MS COCO” and “Pascal VOC)?

2. The content of paper must not be ambiguous related to detection: there are also a classification? In this case how many classes, and what was the accuracy and sensitivity  of the new proposed structure?

3. How was used practically the Pfire in equation (4)?

4. What are the mathematics fundaments for Average Precision (eq. 14) end why they are not used know measure for quality of classification?

5. A question arise from results, that is the measures of classification are comparable for other structures when are taken into account the table 4 and 5? More clearly, at the same precision and accuracy of classification, e.g., the "Detection time" is better in comparison with Tiny-YOLO?

Author Response

Point 1: The abstract must be clear: what dataset was used to test the ReSTiNet, the new proposed architecture (probably MS COCO” and “Pascal VOC)?

Response 1: The authors would like to thank the reviewer for his concern. In response to the reviewer's concerns, the authors would like to state that this study uses the MS COCO and Pascal VOC datasets. The authors have also modified the abstract to make it clearer.

Point 2: The content of paper must not be ambiguous related to detection: there are also a classification? In this case how many classes, and what was the accuracy and sensitivity of the new proposed structure?

Response 2: The authors would like to thank the reviewer for the concern. This article focuses on only the detection of human class, and no classification task is performed for other classes from the dataset. The experiments are performed on human detection, a subset of object detection. In this study, the "MS COCO" and "Pascal VOC" datasets are utilized. The "Pascal VOC" dataset includes "Pascal VOC 2007" and "Pascal VOC 2012".  8540 images of human objects from "Pascal VOC" and 45,174 images of human objects from "MS COCO-train2014" were utilized for this experiment. The two datasets are split 80/20 for training and validation, respectively. When calculating mAP values for both datasets, the IOU ("intersection over union") is set by default to 0.5. 1208 images from the "INRIA" dataset are used to evaluate the recognition speed of the proposed ReSTiNet model. Only human objects were extracted from these two datasets. We separate images containing only a single human subject. Please refer to section 4.2: Data-Set Specification for the details.

Point 3: How was used practically the Pfire in equation (4)?

Response 3: The authors would like to thank the reviewer for the concern. The fire equation is used to reduce the parameters of the fire module used in the network. Equation 4 is the concatenated version of the squeeze and expand portion of the module. The squeeze portion reduces the convolutional layer from 3x3 to 1x1, and in the expanded portion, both 1x1 and 3x3 layers are used. Then, with the fire equation, the two portions are merged. Table 6 shows the practical output from equation 4 in terms of calculating the number of parameters.

Point 4:  What are the mathematics fundaments for Average Precision (eq. 14) end why they are not used know measure for quality of classification?

Response 4: The authors would like to thank the reviewer for the concern. We have updated the equation no. 14. Please see the updated manuscript, line no. 447-452.

Point 5: A question arise from results, that is the measures of classification are comparable for other structures when are taken into account the table 4 and 5? More clearly, at the same precision and accuracy of classification, e.g., the "Detection time" is better in comparison with Tiny-YOLO?

Response 5: The authors would like to thank the reviewer for the concern. We compared the developed ReSTiNet model with other lightweight models. In our experiment, we found that ReSTiNet outperformed MobileNet and SqueezeNet in terms of mAP. This article focuses on only the detection of human classes, and no classification task is performed for other classes from the dataset. The experiments are performed on human detection, a subset of object detection.  Please check section 4.6. (ReSTiNet performance comparison with other lightweight methods), line no. 487-500. ReSTiNet’s detection time is better than Tiny-YOLO while maintaining a higher mAP.

Round 2

Reviewer 1 Report

The revision has addressed all my concerns.

Author Response

The revision has addressed all my concerns.

  • The authors would like to thank the reviewer for his tremendous and generous efforts in reviewing the article, which helped the authors to improve the quality of the manuscript.

Reviewer 2 Report

This paper proposes a lightweight CNN architecture and improves the performance of the baseline models. I think the authors have sufficiently provided revisions based on previous comments. Here, I just have some minor comments:
1. In detection results presented by Figures 6 and 7, the confidence values are not readable. Please, enhance/enlarge them (numbers in the bounding boxes) to improve their readability.
2. As we can see, the mAP performance of proposed method achieved about 64%; This means that there is still room for improvement. Therefore, please show some bad results too, i.e. cases that the proposed method could not handle with.

Author Response

This paper proposes a lightweight CNN architecture and improves the performance of the baseline models. I think the authors have sufficiently provided revisions based on previous comments. Here, I just have some minor comments:

Point 1: In detection results presented by Figures 6 and 7, the confidence values are not readable. Please, enhance/enlarge them (numbers in the bounding boxes) to improve their readability.

Response 1:

  • The authors wish to thank the reviewer for his valuable efforts in guiding the authors in improving the manuscript. At the reviewer's suggestion, Figures 6 and 7 have been updated. The enlarged figure was included in the revised manuscript. However, due to the page size and size ratio of the figures, the images may still not be clearly visible in the 100% view of the PDF file. To compensate for this problem, the images are hosted online and the links to the original images are provided below and also mentioned in the manuscript.
  • Image links:
  • Figure 6: https://i.ibb.co/6FhDYf5/P1-comp.png  
  • Figure 7: https://i.ibb.co/tDs6xPB/P2-comp.png

Point 2: As we can see, the mAP performance of proposed method achieved about 64%; This means that there is still room for improvement. Therefore, please show some bad results too, i.e. cases that the proposed method could not handle with.

Response 2: The authors would like to thank the reviewer for the insightful review. Figure 7(a) shows the scenario with which the proposed method faced problems, as suggested by the reviewer. This scenario shows that the proposed method has a problem in any dense scenario. However, the detection is better than the detection of Tiny-Yolo, but there is an obvious room for improvement in the proposed method. The discussion can be found in lines 548-554.

Reviewer 3 Report

All the requirements are fulfilled. The paper can be published.

Author Response

All the requirements are fulfilled. The paper can be published.

  • The authors would like to thank the reviewer for his tremendous and generous efforts in reviewing the article, which helped the authors to improve the quality of the manuscript.
